# Developing an Instrument to Measure Self-Efficacy, Challenges and Knowledge in Oral Care among Geriatric Home Care Nurses—A Pilot Study

**DOI:** 10.3390/ijerph181910019

**Published:** 2021-09-23

**Authors:** Taru Aro, Marja-Liisa Laitala, Hannu Vähänikkilä, Helvi Kyngäs, Antti Tiisanoja, Anna-Maija Syrjälä

**Affiliations:** 1Research Unit of Oral Health Sciences, Faculty of Medicine, University of Oulu, 90014 Oulu, Finland; marja-liisa.laitala@oulu.fi (M.-L.L.); antti.tiisanoja@oulu.fi (A.T.); anna-maija.syrjala@oulu.fi (A.-M.S.); 2Medical Research Centre Oulu, Oulu University Hospital, University of Oulu, 90014 Oulu, Finland; 3Northern Finland Birth Cohorts, Arctic Biobank, Infrastructure for Population Studies, Faculty of Medicine, University of Oulu, 90014 Oulu, Finland; hannu.vahanikkila@oulu.fi; 4Research Unit of Nursing Science and Health Management, Oulu University Hospital, 90014 Oulu, Finland; helvi.kyngas@oulu.fi; 5Periodontology and Geriatric Dentistry, Research Unit of Oral Health Sciences, University of Oulu, 90014 Oulu, Finland

**Keywords:** geriatric nurses, oral hygiene, self-efficacy, knowledge, mixed-method study, older people

## Abstract

The role of geriatric nurses is essential in preventing oral health problems of older people with impaired daily functioning. Nurses have reported low self-efficacy with regard to oral health care practices and wish to receive more information on the topic. The main aim of this pilot study was to develop an instrument to measure the self-efficacy beliefs, challenges and knowledge of geriatric home care nurses with regard to the oral health care of older. A questionnaire was developed to evaluate geriatric home care nurses’ self-efficacy beliefs, challenges and knowledge regarding the oral health care of the older people. In this case, 18 nurses participated in a one-day intervention and filled in the questionnaire before and after the intervention. The comments and questions of the nurses were analysed utilising the principles of inductive content analysis. Cronbach’s alpha for the scales varied between 0.69–0.79. The interactive intervention improved both nurses’ self-efficacy beliefs and oral health-related knowledge, and most of the challenges faced by nurses in older people’s oral health care were diminished. Financial resources and older people’s self-determination were the most common limitations to oral care. In this pilot study, we developed an instrument to measure geriatric home care nurses’ self-efficacy beliefs, challenges and knowledge regarding older people’s oral health care. In the future, this instrument can be validated with a larger study population.

## 1. Introduction

Over the past few decades, life expectancy has increased constantly, especially in industrialised countries. In the European Union (EU), the number of people aged 80 and older was 27.3 million in 2016, compared to 7 million fewer in 2006 [1]. Older people often suffer from periodontitis, dental caries, dry mouth and mucosal problems as well as problems with chewing and removable dentures [2,3]. Poor oral health has a negative impact on the quality of life [4] and is reported to increase the risk for systemic diseases, such as cardiovascular and neurodegenerative disorders, rheumatoid arthritis [5], chronic obstructive pulmonary disease (COPD), cancer and diabetes [6] and problems related to nutrition [7]. In addition to chronic disorders, poor oral health increases the risk for severe acute diseases, e.g., aspiration-associated pneumonia [8]. Despite the abovementioned risks, the oral health care of older people in nursing homes is still often overlooked [9].

The role of nurses is essential in preventing oral health problems among older people with impaired daily functioning. Today, about 73,000 older people in Finland receive regular services from geriatric home care nurses, which enables them to live at home as long as possible [10]. Among other daily activities, home care nurses should also help with oral self-care practices and arrange regular dental visits.

In order to manage oral health practices successfully, nurses need knowledge regarding oral diseases among older people, the skills to perform daily oral health practices and self-efficacy [11,12] in oral health care in order to overcome the various challenging situations faced when caring for geriatric home care clients [12]. It has been reported previously that nurses’ and caregivers’ knowledge about oral health care of older people is generally moderate to high [13,14] but self-efficacy with regard to the oral health care of older people is limited [15,16]. Furthermore, the challenges reported by nurses in oral care in nursing homes are older people’s limitations in cooperation, low priority of oral care and lack of time and lack of oral care instruments [17,18,19,20]. Most of the studies on oral health care provided by nurses focus on nursing staff working in nursing homes. Recently, an instrument to measure nurses’ self-efficacy and attitudes toward providing oral health care in nursing homes was developed in the US [21]. The oral care challenges could be different among older people living independently in their own homes, whether or not they receive regular services from geriatric home care nurses [22].

In our previous reports, we found that geriatric home care nurses had low self-efficacy with regard to daily oral health care practices, and they wished to receive more information and education about oral health issues [15,16]. The main aim of this pilot study was to develop an instrument explicitly for geriatric home care nurses to analyse self-efficacy beliefs, challenges and knowledge regarding the oral health care of older clients. A secondary aim was to pilot the impact of an interactive oral health intervention on the self-efficacy beliefs and challenges in oral health care and on oral health-related knowledge of geriatric home care nurses and to analyse the comments of the participating nurses using a qualitative approach.

## 2. Materials and Methods

### 2.1. Study Population and Collection of Data

Our study population was comprised of geriatric home care nurses working in the homes of older people in the region of Soite Primary Care Services in Mid-Western Finland (https://www.soite.fi/, accessed 23 September 2021). Soite provides public primary and special health care services (including oral health care) as well as social services in a province that is home to 78,000 people. Soite was chosen as a convenient sample because systematic oral health education had not been previously reported there for nurses and there was a recognised need for this education. The study was conducted in September 2019 in two of Soite’s municipalities. All geriatric home care nurses (*n* = 42) working in these regions were invited to participate by an e-mail sent to the head nurses. Altogether 43% (*n* = 18) of the nurses were able to participate in the study because there is a demand for an adequate number of nurses to always be on duty due to the nature of this health care field.

### 2.2. Questionnaire

The questionnaire was developed by A-MS and M-LL and consisted of nurses’ self-efficacy beliefs, challenges and knowledge regarding older people’s oral health and oral health-related disorders. Furthermore, background factors were inquired. Before the study, the questionnaire was piloted with home care nurses in the city of Oulu and considered to be clear and unambiguous. The Cronbach’s alpha for the self-efficacy scale in the pilot study was 0.71, indicating high internal consistency.

#### 2.2.1. Background Factors of the Participants

The age, educational background, the time since graduation and the time working as a geriatric home care nurse were asked. In addition, there were questions about the nurses’ oral health practices when caring for older people and the nurses’ own oral health care habits (tooth brushing, flossing, dental visits).

#### 2.2.2. Self-Efficacy Beliefs (10 Items)

Self-efficacy refers to how confident a person is in performing specific actions in order to achieve a certain goal [11]. The self-efficacy items were formulated by A-MS and M-LL. The content validity of the self-efficacy items was based on a review of Burrell [23], who proposed that by asking “I am confident that…” is a valid measure of self-efficacy. The items consist of home care nurses’ essential everyday practices in oral health care among older people and are valid in the work of geriatric home care nurses. In this section of the questionnaire a 10-point scale was used, with a score of 0 meaning “I’m not at all confident” and 10 meaning “I’m completely confident”.

#### 2.2.3. Nurses’ Challenges in Oral Health-Related Practices among Older People (7 Items)

All items included possible barriers to oral health care among clients encountered by home care nurses. The items were partially based on our previous study [16]. There were items concerning older people, such as co-operation, and items concerning the nurses and their work, such as lack of time, knowledge or materials and nurses’ disgust with dentures and dirty mouths that produce bad odours. All of the statements on challenges began with “The oral health care of the older people is challenging because…”. A 10-point scale was used, with a score of 0 meaning completely disagree and 10 meaning completely agree.

#### 2.2.4. Oral Health-Related Knowledge (9 Items)

Items about oral health care-related knowledge evaluated knowledge about the etiological factors of oral diseases and specific geriatric problems. These items were partly based on the previous questionnaire [24]; new items were also added, including items about current knowledge of the association between oral health and geriatric diseases [6,8]. The knowledge items were formulated on a 5-point Likert scale (completely agree, agree, no idea, disagree and completely disagree).

### 2.3. Structured Intervention Event to Support Nurses in Oral Health Care

In the beginning of the one-day intervention, nurses were asked to complete the questionnaire. After that, a lecture was given, followed by hands-on training in oral hygiene practices. At the end of the day, the nurses filled in the same questionnaire again.

The lecture, which incorporated a Power point-presentation, included themes on oral health and the most common oral problems of older people (dental caries, periodontal diseases, denture stomatitis and oral cancer) and the association between oral and systemic diseases. At the end of the lecture, the lecturer used a model to demonstrate how to clean older people’s teeth and removable dentures. At the beginning of the hands-on training, two of the authors demonstrated cleaning of the teeth, mouth and oral mucosa in practice. M-LL acted as the patient and TA performed the role of the nurse. In the hands-on training, the nurses were asked to select a pair and they had the opportunity and were encouraged to clean each other’s mouths and teeth under the guidance of the research group.

The intervention centred around oral health topics was designed by three of the authors: a DDS, PhD and experienced specialist in gerodontology (A-MS); a DDS, PhD and experienced specialist in public dental care (M-LL); and a DDS, PhD student (TA) from the University of Oulu, Finland. The same three authors carried out the intervention day. The group met before the intervention and thoroughly evaluated the schedule, content and details of the day (Table 1). The group agreed on the role of each member during the day and how to deal with possible challenges.

During the whole intervention day, nurses had the possibility to ask about older people’s oral health care and practices. The aim was to create an atmosphere as informal as possible; in small groups (9 people in each group), nurses had enough courage to ask about and comment on oral health-related issues. The comments and questions of the participating nurses were also recorded by writing them down during the day.

### 2.4. Psychological Models

Our aim in the intervention was to strengthen nurses’ self-efficacy. We utilised self-efficacy sources presented by Bandura [11], modified for this context. Self-efficacy consists of four major sources: performance accomplishments, vicarious experience, verbal persuasion and emotional arousal [11]. These four sources are all connected to and enhance each other.

Performance accomplishment means that a person’s repeated success creates strong efficacy expectations. Occasional failures can strengthen motivation if the person finds through experiences that even the most difficult obstacles can be handled as well. Vicarious experience means that a person notices somebody else succeeding in a particular matter, and the success of this somebody else strengthens his/her own belief in succeeding in the same matter, which encourages the person to persist with his/her efforts. By verbal persuasion, used by health care professionals, for example, a person is persuaded to believe he/she can handle difficult situations. Emotional arousal can be harmful to one’s perceived potency in challenging situations.

In the hands-on-course, nurses were trained in special skills related to oral care practices, which boosts their performance accomplishment. Verbal persuasion was used, and nurses were advised to perform oral health care practices on older people. Nurses were shown how to clean removable dentures, brush the teeth and clean the oral mucosa, and they were asked to clean each other’s mouths; they could thus see others coping with these situations, and this helped them manage with oral health care practices with clients. This enhanced the vicarious experience. We tried to create an atmosphere at the intervention that was as informal as possible, so that it would not create any fear of oral health care, which is one of the points in emotional arousal.

### 2.5. Statistical Methods

Descriptive statistics were presented as means and standard deviations for continuous variables and proportions for categorical variables. Cronbach’s alpha was used to measure the reliability and internal consistency of the scales for self-efficacy beliefs, challenges and oral health-related knowledge. The sum scores of the answers for the scales of self-efficacy beliefs and challenges in oral health care and the proportions of answers in oral health-related knowledge scale were calculated before and after intervention. A paired samples *t*-test was used to compare mean sum scores of the scales for self-efficacy beliefs and oral health-related knowledge before and after the intervention. For the analyses, we used IBM SPSS Statistics version 26.0 for Windows (Chicago, IL, USA).

### 2.6. Qualitative Analysis

Open-ended data (nurses’ comments and questions) were analysed by using the principles of inductive content analysis [25]. Themes were created by grouping together the open codes with similar content and named according to their content.

### 2.7. Ethical Consideration

Participation was voluntary and data were collected and analysed without participants’ IDs. The ethical committee of the Hospital District of Northern Ostrobothnia approved the study (EETMK: 59/2019 172).

## 3. Results

Altogether 18 nurses participated in the intervention study. A description of the study participants is shown in Table 2. Most of the participants were 50–59 years old and had more than 14 years since graduation and more than 14 years’ experience working in geriatric home care.

### 3.1. Self-Efficacy

Cronbach’s alpha was 0.79 for the self-efficacy scale, indicating high internal consistency and reliability of the self-efficacy scale. The mean scores for the self-efficacy items before and after the intervention are shown in Table 3. The mean sum score for the total scale was 55.2 (SD 8.3) before the intervention and 68.9 (SD 13.4) after the intervention (paired samples *t*-test, *p* = 0.001) and the intervention improved self-efficacy perception in all dimensions of the self-efficacy scale. The greatest change in mean scores was in the statement: “I am confident that I am able to notice mucosal inflammation related to older people’s use of removable dentures”, with a mean score of 4.5 (SD 1.6) before the training and 7.2 (SD 1.7) after it.

### 3.2. Challenges

Cronbach’s alpha was 0.72, indicating high internal consistency and reliability for the scale of challenges. The mean scores for each item before and after the training are shown in Table 4. The most noticeable diminished challenge was in the statement: “The oral care of older people is challenging because I don’t have enough knowledge of how to clean older people’s mouths” (mean score 5.8 before the education and 2.7 after the education). The challenge also decreased for the statement: “The oral health care of older people is challenging because I can’t evaluate what the problem is in older people’s mouths” (mean score 6.2 before the education and 4.2 after it). There was an increase in the challenge described by the statement “The oral care of the older people is challenging because older people do not have decent oral cleaning instruments, such as a toothbrush and toothpaste” (mean score 5.3 before the education and 6.1 after it).

### 3.3. Oral Health-Related Knowledge

In order to improve Cronbach’s alpha, we excluded the item “I have received enough information about the factors that affect oral health”, which did not describe any detailed knowledge about geriatric oral care. After that, Cronbach’s alpha was 0.69. Knowledge was increased noticeably after the intervention. The mean sum score for the total scale was 28.9 (SD 3.3) before the intervention and 32.5 (SD 1.3) after the intervention (paired samples *t*-test, *p* = 0.001). The percentages of answers related to oral health-related knowledge before and after the intervention are shown in Table 5.

### 3.4. Qualitative Analysis of the Participants’ Comments

The raw data was read thoroughly, and all the comments and questions were reduced into open codes. The open codes were grouped into main themes according to the principles of inductive content analysis [25,26]. Altogether six main themes emerged in our study: knowledge, skills, resource challenges, solutions, older peoples’ self-determination and self-empowerment, and the habits of older people. Simplified expressions of the themes are shown in Table 6.

#### 3.4.1. Knowledge

Most of the comments and questions related to the participants’ knowledge were about removable dentures, such as loose-fitting dentures and their cleaning. The oral health of older people was thought to be poor: a couple of the nurses mentioned that “the oral health status of older people is so poor that the geriatric home care nurse cannot do anything”. One of the nurses asked “whether a patient with memory disorders would allow a dentist to fill in a cavity”. The inability to perform daily oral health care practices was also acknowledged, as one registered nurse commented that “the importance of oral care was emphasised in the nursing studies, but practical training was lacking”.

#### 3.4.2. Skills

Comments related to skills were also mostly about removable dentures. Nurses asked how to clean dentures with persistent plaque. One nurse also asked “if the dentures cause pressure on the mucosa, what can a nurse do if the dentist does not have time for an appointment?” The difficulty of cleaning of older people’s own natural teeth was also noticed as one nurse mentioned “it is easier to take care of dentures than to clean their own teeth”.

#### 3.4.3. Resource Challenges

Lack of financial resources was revealed in the open comments, as the nurses mentioned that “there is less money to spend in the countryside compared to larger towns”. One nurse also stated that “older people do not have enough money for dental services”. Furthermore, it was commented that “people living in the countryside are not used to using services like dental services”. A lack of time for oral care was also noted as it was mentioned that “nurses do not have enough time to take care of oral health during home visits”.

#### 3.4.4. Solutions

Most of the nurses commented that this education prompted them to take care of older people’s oral health. It was also noted that oral health care should be included in a daily treatment plan. The regularity of dental appointments was mentioned by one nurse: “older people should get a regular call to a dentist, like children do [in Finland]”. The nurses also noted their own responsibility as some of the nurses commented that “it is the responsibility of the nurse to make an appointment with a dentist”.

#### 3.4.5. Older People’s Self-Determination and Self-Empowerment

Nurses mentioned that older people living at home have greater self-determination than older people living in care facilities. Older people want to take care of their own oral health care by themselves. “Most of the older people take care of their own oral health care, so I do not bother to help”, one nurse commented.

#### 3.4.6. Habits of Older People

“Older people are accustomed to using particular tools for oral self-care, so it is difficult to change their habits”, one nurse mentioned. Many of the older people are not used to using services, such as visiting a dentist.

## 4. Discussion

Our main result was successfully developing an instrument for geriatric home care nurses to analyse their self-efficacy beliefs, challenges and knowledge related to the oral health care of older clients. Furthermore, the intervention improved nurses’ self-efficacy and oral health-related knowledge and diminished the challenges encountered in the oral health care of older clients. In the qualitative part of the study, as the main limitations in their work, the nurses mentioned older people’s lack of financial resources for dental care and older people’s self-determination and nurses’ lack of time for oral health care practices during home visits. These results give a hopeful outlook for the utilisation of this instrument for measuring nurses’ self-efficacy beliefs, challenges and knowledge in older people’s oral health care.

We developed a self-efficacy scale for nurses, and it showed high internal consistency and reliability (Cronbach’s alpha 0.79). The content validity of our self-efficacy items was good because the formulation of items was based on the review by Burrell et al. [23], which showed that it is valid to measure self-efficacy starting with the statement “I am confident that…”. The face-validity of the items was also good as the statements describe nurses’ essential everyday practices in oral health care among older people and are valid in the work of geriatric home care nurses.

Nurses need high self-efficacy in oral health care in order to overcome various challenging situations among clients. In order to improve nurses’ self-efficacy during the intervention, we utilised Bandura’s theories of self-efficacy [11,12], particularly the sources of performance accomplishment, vicarious experience and verbal persuasion. Some nurses practiced oral health care procedures (toothbrushing, cleaning the oral mucosa) in the mouths of their colleagues while some nurses practiced with models; both enhanced the performance accomplishment of the nurses. The nurses were also shown how to clean the mouth with models and in practice, which enhanced vicarious experience. The mean scores for all self-efficacy items increased after the education, which means that this kind of education may be useful in improving nurses’ self-efficacy beliefs in oral care among older clients.

In the scale of challenges, the face validity of the items was good in that all the items included possible barriers to oral care that home care nurses may encounter in their everyday practices among home dwelling older clients. The content validity of the challenges-related items is supported by the fact that home care nurses have reported some of these same challenges in our previous qualitative study [16], and similar challenges have been previously reported [19]. Furthermore, the scale showed high internal consistency (Cronbach’s alpha 0.72) which supports the reliability of the scale to be used for the present purpose. One of the noteworthy results during the intervention day was that the nurses found that the item concerning oral health care equipment for older people to be one of the more challenging ones. A possible explanation is that the nurses found it very difficult to update older people’s outdated oral care equipment. In addition, in the item “The oral care of older people is challenging because I haven’t got enough time for it” the challenge increased. This is in line with our previous reports [16] and an explanation for the increased challenge may be that after the education, the nurses noticed that they really should focus on the oral health care of older people and it takes more time than they now use during home visits. Nurses have reported a lack of time for oral care in other studies as well [19,27].

Items in the scale of oral health-related knowledge include basic knowledge needed to understand the importance of oral health among older clients, which confirms the face validity of the scale. The content validity of the scale is supported by the fact that the knowledge items were partly based on the previous instrument to measure knowledge about oral health care [24], and there are also items concerning current knowledge about the associations between oral health and general health. These items included parallel items used previously to measure knowledge [6,8]. Furthermore, the knowledge scale showed moderate internal consistency (Cronbach’s alpha 0.69) which supports the reliability of the scale.

During the intervention day, nurses’ knowledge about the oral health care of older people increased noticeably. The increase in knowledge was probably mostly influenced by the lecture. In addition, the informal and relaxed atmosphere during the interactive intervention might have increased knowledge by decreasing the nurses’ threshold for questions and comments. The hands-on training may have improved knowledge about actual daily oral health care practices. It has been found that group discussions and repetition of information, along with clarification of information, improve the behaviour of nurses in the field of oral health care and, consequently, the oral health of older people [18]. There should be a study on whether these kinds of interventions of nurses really improve the actual outcome, i.e., the oral health of older people.

In the qualitative part of our study, the nurses mentioned a lack of resources, such as limited time for home -visits and insufficient financial resources, as barriers to oral care. The compliance and self-determination of older people were also mentioned as barriers. Our results are concomitant with previous studies, in which nurses in nursing homes have been reported as facing various, nearly identical barriers to ensuring the oral hygiene of their older clients, such as heavy workload, lack of skills and knowledge concerning oral care, respect for older people’s self-determination, and an inability to observe their clients’ oral state [28]. The nurses in our study also presented solutions to improving oral care among clients, such as regular dental visits and including oral health care practices in the clients’ daily treatment plan as also previously reported [20,29]. Oral health care practices in the clients’ daily treatment plan would remind the nurses every day about the importance of oral health care.

When comparing older people living at nursing homes to those who live in their own homes, the greater self-determination of older people living at home is a limitation for nurses in trying to help them with their oral health care [19]. The self-determination of home-living older people is more pronounced in terms of their attitude to oral care, use of finances, choosing home care equipment, receiving help in oral care from home care nurses and the visiting dentist when comparing to older persons living in a nursing home where the departments have their own protocols for oral health care. It should be noted that residents who resist have been reported to be one of the most significant barriers to oral care in nursing homes [19] and a Mouth Care Without a Battle approach has been tested in nursing homes recently [21]. In contrast, the nurses in our study did not report resisting oral care by home dwelling older people to be an essential factor. To our knowledge, interventions to improve the self-efficacy of home care nurses in oral health practices among geriatric patients living at home have not been reported and previous reports mainly concern nursing homes [18]. This kind of structured intervention for small groups seems to be effective at improving the self-efficacy of geriatric home care nurses and it is practically impossible to gather all home care nurses to a training event at the same time.

There are very few self-efficacy measures concerning older people’s oral health care and these measures are for staff in nursing homes [15,17,21,30]. Wretman et al. [21] have developed a psychometrically validated measure with factor validity, construct validity and criterion validity. Our self-efficacy scale is targeted explicitly for home care nurses and it showed high internal consistency and reliability, and sufficient content validity and face validity. Due to the small number of participants and no clinical examination of the clients in the study protocol, factor analysis, construct validity and criterion validity could not be analysed. In the future, our measure could be further improved by analysing it with larger study populations, which enables more accurate psychometric validation. Moreover, the oral status of clients, including dental plaque and gingival inflammation, could be analysed in order to evaluate criterion validity.

A strength of this study was the versatile intervention, which contained both information about oral diseases and oral problems experienced by older people and practical training in oral hygiene. Another strength of the study was that we had experienced professionals planning and leading the intervention. The small intervention groups were also found to be beneficial for the intervention as small intervention groups were essential for achieving an atmosphere suitable for an interactive intervention. Qualitative analyses provided important information that could not be obtained with structured questionnaires. In Soite, oral health care education for the geriatric home care nurses had been scarce and non-systematic in the past, and thus, generalising of the results should be carried out with caution. This can be considered also as a limitation of the study. It must be kept in mind that social desirability might have influenced the nurses’ answers in certain questions, and this could have influenced the responses [31]. Furthermore, total anonymity was not achieved as participants filled in the questionnaire during the intervention day in the presence of the researchers, albeit the names of the nurses were not included in the questionnaire.

### Implications

In this pilot study we have presented an instrument to measure the self-efficacy beliefs, challenges and knowledge about the oral health care of older people among geriatric home care nurses. The results are promising and support the reliability and validity of the scales. In the future, these scales should be tested with a larger study population to further confirm the reliability and validity of the scales. We also found that an interactive intervention has a promising effect on nurses’ self-efficacy beliefs and oral health-related knowledge and diminishes challenges in oral health care. In the future there should be a study on whether this kind of intervention actually improves the daily oral health practices of the home care nurses and as an important aim, the actual oral health of older people.

## 5. Conclusions

It can be concluded that our pilot study shows promising results for the instrument developed to measure nurses’ self-efficacy beliefs, challenges and knowledge related to the oral health care of older clients. A short interactive intervention may improve nurses’ self-efficacy beliefs and oral health-related knowledge and diminish challenges in oral care, as found in this pilot study.

## Figures and Tables

**Table 1 ijerph-18-10019-t001:** Schedule of the intervention.

Time	Content	Details
5 min	Introduction	Welcome, the aim of the day, presentation of the participants
15 min	Filling in the questionnaire	
30–45 min	Lecture with Power Point -presentation	Basic oral problems of older people, demonstration with a model for how to clean older people’s teeth and removable dentures.
20 min	Coffee break	
45–60 min	Hands-on training in pairs and interactive discussion	Two of the authors demonstrated cleaning of the mouth and oral mucosa, toothbrushing, dental flossing and the use of interdental brushes.
15 min	Filling in the questionnaire	
	Reflection on the session	Comments and opinions of the participants

**Table 2 ijerph-18-10019-t002:** Characteristics of the participants.

Variable		*n*	%
Age (years)	20–29	3	16.7
	30–39	1	5.6
	40–49	3	16.7
	50–59	10	55.6
	≥60	1	5.6
Basic education	Secondary level (high/vocational school)	14	77.8
	Tertiary level	3	16.7
	No answer	1	5.6
Nursing education	Assistant nurse	9	50.0
	Registered general nurse	5	27.8
	Assistant without degree in nursing	2	11.1
	No answer	2	11.1
Time since graduation (years)	<5	2	11.1
	5–9	3	16.7
	10–14	2	11.1
	>14	11	61.1
Working experience in geriatric home care (years)	<5	2	11.1
	5–9	3	16.7
	10–14	2	11.1
	>14	11	61.1

**Table 3 ijerph-18-10019-t003:** Mean scores for the self-efficacy items before and after intervention.

Proposition: I Am Confident that I…	Before Mean (SD)	After Mean (SD)
…know how to brush older people’s teeth	6.6 (1.7)	8.2 (2.2)
…know how to clean older people’s removable dentures	7.5 (1.5)	8.9 (1.3)
…am able to notice dry mouth of older people	6.4 (1.6)	8.1 (1.4)
…am able to notice cavities in older people’s teeth	4.1 (1.9)	6.2 (2.3)
…am able to notice older people’s inflamed gums	5.4 (1.6)	7.6 (1.7)
…am able to notice mucosal inflammation related to older people’s use of removable dentures	4.5 (1.6)	7.2 (1.7)
…brush older people’s teeth, unless I’m pressed for time	5.0 (2.4)	5.4 (3.1)
…clean older people’s mouth even if co-operation with them is problematic	4.3 (2.1)	4.6 (2.4)
…look at older people’s mouth every day	3.4 (2.4)	3.8 (3.0)
…make an appointment with a dentist if there is a problem in older people’s mouth	8.1 (2.6)	8.9 (1.6)

**Table 4 ijerph-18-10019-t004:** Mean scores for challenges items before and after intervention.

Proposition: The Oral Care of Older People Is Challenging Because…	Before Mean (SD)	After Mean (SD)
…I haven’t got time enough for it	5.6 (1.8)	6.1 (3.0)
…I haven’t got enough knowledge of how to clean older people’s mouths	5.8 (1.5)	2.7 (2.7)
…older people don’t let me look into their mouths	6.6 (2.2)	6.1 (1.4)
…I am disgusted by dirty removable dentures	2.0 (2.4)	2.1 (2.4)
…I am disgusted smelly, dirty mouths	3.2 (2.8)	2.9 (2.8)
…I can’t evaluate what the problem is in older people’s mouths	6.2 (1.8)	4.2 (2.2)
…older people do not have decent oral cleaning instruments such as a toothbrush and toothpaste	5.3 (3.2)	6.1 (2.3)

**Table 5 ijerph-18-10019-t005:** Nurses’ (*n* = 18) responses (in percentages) to the items of oral health-related knowledge before and after intervention.

Proposition: The Oral Care of Older People Is Challenging Because…		Completely Disagree (%)	Disagree (%)	No Idea (%)	Agree (%)	Completely Agree (%)
I have received enough information about the factors that affect oral health	before	11.1	33.3	27.8	27.8	0
	after	0	0	0	22.2	77.8
Oral health has an impact on general health	before	0	0	0	5.6	94.4
	after	0	0	0	0	100
It’s not normal that gums bleed while brushing teeth	before	0	0	5.6	16.7	77.8
	after	0	0	5.6	11.1	83.3
Bad breath depends on the bacteria in the mouth	before	0	16.7	0	61.1	22.2
	after	0	0	0	16.7	83.3
Poor dental hygiene increases the risk of pneumonia	before	0	0	5.6	50.0	44.4
	after	0	0	0	0	100
Gingivitis depends on the bacteria in the mouth	before	0	5.6	5.6	61.1	27.8
	after	0	0	0	5.6	94.4
Repeated usage of sugar increases the risk of dental caries	before	0	5.6	0	11.1	83.3
	after	0	0	0	0	100
Long-lasting inflammation in the mouth can increase the risk of memory disorder	before	0	0	56.6	16.7	27.8
	after	0	0	0	16.7	83.3
Tooth loss is not normal when aging	before	0	5.6	11.1	50.0	33.3
	after	5.6	11.1	0	16.7	63.1

**Table 6 ijerph-18-10019-t006:** Open codes of themes.

Knowledge	Skills	Resource Challenges	Solutions	Older Peoples’ Self-Determination and Self-Empowerment	Habits of Older People
Appearance of oral cancer	Plaque removal from removable dentures	Lower education level in countryside	Education increased nurses’ awareness	Older people want to take care of themselves	Not used to using services
Cleaning removable dentures	Removing removable dentures from mouth	Less money for services	Regular oral health examination for older people	Living at home and self-determination	Not used to oral health care
First aid when dentures cause pressure	First aid when removable dentures cause pressure	Nurses do not have enough time	Oral health care in daily treatment plan	Older people take care of their own mouths	Cleaning of own, old removable dentures
Dementia and filling a tooth cavity	Easier to clean removable dentures than natural teeth	Nurse cannot do anything when an older person’s mouth is in bad condition	Education to be a necessity	Older people are used to using specific tools for oral hygiene	Older people feel that it is useless to go to a dentist
Plaque removal from removable dentures			Nurse encourages client to book an appointment with a dentist		
Removing removable dentures from mouth			Practical training in nursing education		
Scarce oral health education in nursing education					
Nurse cannot do anything when an older person’s mouth is in bad condition					

## Data Availability

The data presented in this study are available on request from the corresponding author. The data are not publicly available due to ethical reasons.

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
