# Peer review of "Developing an Instrument to Measure Self-Efficacy, Challenges and Knowledge in Oral Care among Geriatric Home Care Nurses—A Pilot Study"

_ijerph, 2021, doi:10.3390/ijerph181910019_

Round 1

Reviewer 1 Report

Dear Authors, 

As for the previous rounds, you carried out a thorough and comprehensive review. The only clarifications I need from the previous rounds is the one regarding sample size. Was this approved/discussed with a statistician?

Once clarified this point, I believe the manuscript would be suitable for publication and that it will be an interesting starting point for future studies.

Author Response

As for the previous rounds, you carried out a thorough and comprehensive review. The only clarifications I need from the previous rounds is the one regarding sample size. Was this approved/discussed with a statistician?

Once clarified this point, I believe the manuscript would be suitable for publication and that it will be an interesting starting point for future studies.

  • Thank you for your comments. The research plan, sample size as well as statistical methods used were approved by a statistician (one of the authors). We consider that the sample size is acceptable for this pilot study, but indeed, bigger sample size in future studies can confirm our results.

Reviewer 2 Report

I would like to congratulate the authors of the present manuscript. The pilot nature of this study implies a series of limitations which have been transparently addressed, and the conclusions are in accordance which such limitations. 

I have read the review history and have come to the conclusion that, after the responses to the rest of the reviewers, the main concerns have been addressed and the manuscript's quality has improved substantially. Therefore, I would recommend its acceptance.

My only suggestion would be you further assess the quality of the English language throughout in the present manuscript. A series of uncommon expressions were used, such as "In future", instead of "in the future"; or the use of the word "further" as a "furthermore" or "moreover".

Thank you.

Author Response

I would like to congratulate the authors of the present manuscript. The pilot nature of this study implies a series of limitations which have been transparently addressed, and the conclusions are in accordance which such limitations. 

I have read the review history and have come to the conclusion that, after the responses to the rest of the reviewers, the main concerns have been addressed and the manuscript's quality has improved substantially. Therefore, I would recommend its acceptance.

My only suggestion would be you further assess the quality of the English language throughout in the present manuscript. A series of uncommon expressions were used, such as "In future", instead of "in the future"; or the use of the word "further" as a "furthermore" or "moreover".

Thank you.

  • Thank you for your comments. The text is now re-checked by a professional linguistic reviewer. We have revised the language of the manuscript according to his suggestions.

Reviewer 3 Report

Congratulations to the Authors for this novel and important study!

I have a few  minor comments only:

1) Is the questionnaire in Finnish only?

Do you plan to produce it in English as well?

2) Were the questionnaire you used validated?

3) Do you feel this one-off intervention could be implemented elsewhere to increase awareness of the implicated parties? Perhaps you can make suggestions on where else it could be used.

Congratulations again!

Author Response

Congratulations to the Authors for this novel and important study!

I have a few minor comments only:

1) Is the questionnaire in Finnish only?

Do you plan to produce it in English as well?

  • The questionnaire is in Finnish only. All the questions have been translated into English for this study, but we have not published the English questionnaire.

2) Were the questionnaire you used validated?

  • The questionnaire was not validated before this study. We validated all the scales of questionnaire, as reported in the study.

3) Do you feel this one-off intervention could be implemented elsewhere to increase awareness of the implicated parties? Perhaps you can make suggestions on where else it could be used.

  • This kind of intervention would be implemented for example as on-the-job-training among geriatric home care nurses in their meetings and staff training days. As mentioned in implications, this kind of intervention should be tested also with a larger study population.

This manuscript is a resubmission of an earlier submission. The following is a list of the peer review reports and author responses from that submission.

Round 1

Reviewer 1 Report

This manuscript described formulating an instrument to measure self-efficacy beliefs, challenges and knowledge in geriatric home care nurses. Oral diseases can be prevented in advance stage but due to lack of dental care knowledge and confidence in ability to manage oral diseases in geriatric home care nurses, older people may experience serious dental problems which could negatively affect the overall quality of life. This study signifies the importance of training and assessment in geriatric home care nurses to improve oral health in older people. The study is well designed but there are some limitations.

  1. Although this is a pilot study, the sample size is still very small to derive a firm conclusion.
  2. Since, person’s intention to carry out a behavior and their perceived capability to carry out a behavior remain separate, characteristics of the participants given in table 2 is very important. Considering that, authors should categorize participants based on age, nursing education and working experience to measure self-efficacy, challenges and knowledge.
  3. For this study, authors should also present participants data on their orodental care before and after intervention. Knowledge and self-efficacy towards their own orodental care may influence the oral care they give to the elderly patients.
  4. Table 3 should include the proposition of the importance of removing denture at night which can prevent choking.

Author Response

This manuscript described formulating an instrument to measure self-efficacy beliefs, challenges and knowledge in geriatric home care nurses. Oral diseases can be prevented in advance stage but due to lack of dental care knowledge and confidence in ability to manage oral diseases in geriatric home care nurses, older people may experience serious dental problems which could negatively affect the overall quality of life. This study signifies the importance of training and assessment in geriatric home care nurses to improve oral health in older people. The study is well designed but there are some limitations.

  1. Although this is a pilot study, the sample size is still very small to derive a firm conclusion.
    • We have revised conclusions as follows: A short interactive intervention may improve nurses’ self-efficacy beliefs and oral health-related knowledge and diminish challenges in oral care, as found in this pilot study.
  2. Since, person’s intention to carry out a behavior and their perceived capability to carry out a behavior remain separate, characteristics of the participants given in table 2 is very important. Considering that, authors should categorize participants based on age, nursing education and working experience to measure self-efficacy, challenges and knowledge.
    • We agree. However, in this pilot study, the sample size did not allow deeper analysis as regards subgroups.
  3. For this study, authors should also present participants data on their orodental care before and after intervention. Knowledge and self-efficacy towards their own orodental care may influence the oral care they give to the elderly patients.
    • We did not have clinical data on the oral health of the participants. We agree that this would be an interesting issue in future research.
  4. Table 3 should include the proposition of the importance of removing denture at night which can prevent choking.
    • Thank you for your suggestion. In the present questionnaire we did not have this question.

Reviewer 2 Report

Dear Authors,

I would like to thank you and congratulate you the hard work and dedication to this project. This work is very interesting and the structure is original and stimulating; I appreciated the double administration of the questionnaire, before and after the intervention. I have few questions that I would like to have explained.

  • How was sufficient sample size established?
  • Were there any questions not answered from some of the participants? In that case, how did you solve the problem in analyzing the data?
  • In materials and methods you say that 'Soite was chosen as a convenient sample because systematic oral health education had not been previously reported there for nurses and there was a recognised need for education.' It is an interesting explanation, but don't you think that it would be worth mentioning it as limitation of the study as well? Furthermore, a small section regarding limitations of the study should be added at the end of the discussion.
  • The qualitative analysis of the comments of the interviewees is really interesting, such as the reduction in open codes. How does it translates into input and suggestions for future studies? (e.g., adding other questions, adding explanations, tips or instructions to lectures or interventions...)
  • I would consider updating reference 11, it is really dated. Isn't there a more recent and updated alternative?

Best regards

Author Response

I would like to thank you and congratulate you the hard work and dedication to this project. This work is very interesting and the structure is original and stimulating; I appreciated the double administration of the questionnaire, before and after the intervention. I have few questions that I would like to have explained.

  • How was sufficient sample size established?
    • We invited all the geriatric nurses in Soite, but only those who were not on duty during the intervention, had possibility to participate (explained in page 2, lines 82-86).
  • Were there any questions not answered from some of the participants? In that case, how did you solve the problem in analyzing the data?
    • There were only a few questions (altogether less than five) with missing value. As suggested by the expert in statistics, in cases with missing value we calculated the item’s mean score and used it in the analyses.
  • In materials and methods you say that 'Soite was chosen as a convenient sample because systematic oral health education had not been previously reported there for nurses and there was a recognised need for education.' It is an interesting explanation, but don't you think that it would be worth mentioning it as limitation of the study as well? Furthermore, a small section regarding limitations of the study should be added at the end of the discussion.
    • This is true, thank you. We have revised this in discussion (page 11, lines 380-382), as follows: In Soite, previous oral health care education for the geriatric home care nurses had been scarce and non-systematic and thus, generalizing of the results should be carried out with caution. This can be considered also as a limitation of the study.
  • The qualitative analysis of the comments of the interviewees is really interesting, such as the reduction in open codes. How does it translates into input and suggestions for future studies? (e.g., adding other questions, adding explanations, tips or instructions to lectures or interventions...)
    • We agree that qualitative analyses can give important information which is not able to be achieved with quantitative analysis of structured questionnaires. Qualitative approach may help to better understand the nurses’ challenges and everyday work. This is now added in discussion (page 11, line 378-379).
  • I would consider updating reference 11, it is really dated. Isn't there a more recent and updated alternative?
    • Albert Bandura presented social cognitive theory in 1970’s, thus, we would like to keep this reference, but have also added his more recent paper addressing social cognitive means and self-efficacy: Bandura A. Health promotion by social cognitive means. Health Educ Behav 2004; 31: 143-64.

Reviewer 3 Report

Frankly speaking, the study is not based on a novelty idea, even the readers or the finish population can take a greater benefits from the well designed and strongly whieghted study that was recently published  by ( et al., 2020) in that field.

The authors did not even mention it in their references ???

In addition to all of the above, the authors study still in its pilot stage.

Author Response

Frankly speaking, the study is not based on a novelty idea, even the readers or the finish population can take a greater benefits from the well designed and strongly whieghted study that was recently published  by (Wretman et al., 2020) in that field.

The authors did not even mention it in their references ???

In addition to all of the above, the authors study still in its pilot stage.

  • We have now added this interesting report in the introduction (page 2, lines 59-60).
  • The results of this pilot study encourage us to continue research in this field.

Round 2

Reviewer 1 Report

I do not agree fully with the authors. This is an interesting pilot study but can definitely be improved. 

Reviewer 2 Report

Dear Authors, 

Thank you for your thorough review and the effort you put in being comprehensive in every answer. 

I am confident this will be an interesting starting point for future studies.

Reviewer 3 Report

The authors, added the (Wretman et al., 2020) study in their introduction, without any serious discussion to it.

Actually i feel that they can built their study on the Wretman (et al., 2020) works that would safe time and effort to the finish population in that field. 

Methods:

Lines between 112 till 118 does not belong to (Wretman et al., 2020) please do correct that.